# Formation Kinetics Evaluation for Designing Sustainable Carbon Dioxide-Based Hydrate Desalination via Tryptophan as a Biodegradable Hydrate Promotor

Muhammad Saad Khan [1] , Bhajan Lal [1,2,*] , Hani Abulkhair [3,4] , Iqbal Ahmed [3,4] , Azmi Mohd Shariff [1,2] , Eydhah Almatrafi [3,4] , Abdulmohsen Alsaiari [3,4] and Omar Bamaga [3,4]

1   CO2 Research Centre (CO2RES), Universiti Teknologi PETRONAS,
    Bandar Seri Iskandar 32610, Perak, Malaysia
2   Chemical Engineering Department, Universiti Teknologi PETRONAS,
    Bandar Seri Iskandar 32610, Perak, Malaysia
3   Center of Excellence in Desalination Technology, King Abdulaziz University, Jeddah 80200, Saudi Arabia
4   Mechanical Engineering Department, King Abdulaziz University, Jeddah 80200, Saudi Arabia
*   Correspondence: bhajan.lal@utp.edu.my; Tel.: +60-10-3858-473

**Abstract:** Desalination using hydrates is a developing field, and initial research promises a commercially feasible approach. The current study proposes the natural amino acid, namely tryptophan, as a biodegradable gas hydrate promotor for desalination applications to speed up the hydrate formation process. Its kinetic behavior and separation capabilities with $CO_2$ hydrates were investigated. The studies were carried out with varying concentrations (0.5, 1, and 2 wt.%) of tryptophan at different experimental temperatures (274.15, 275.15, 276.15, and 277.15 K) at 3.5 and 4.0 MPa pressure and 1 wt.% brine concentration. The induction time, initial formation rates, gas uptake, and water recovery are characterized and reported in this work. Overall finding demonstrated that tryptophan efficiently acted as a kinetic hydrate promotor (KHP), and increased tryptophan quantities further supported the hydrate formation for almost all the studied conditions. The formation kinetics also demonstrated that it shortens the hydrate induction time by 50.61% and increases the 144.5% initial formation rate of $CO_2$ hydrates for 1 wt.% addition of tryptophan at 274 K temperature and 4.0 MPa pressure condition. The study also discovered that at similar experimental conditions, 1 wt.% tryptophan addition improved gas uptake by 124% and water recovery moles by 121%. Furthermore, the increased concentrations of tryptophan (0.5–2 wt.%) further enhance the formation kinetics of $CO_2$ hydrates due to the hydrophobic nature of tryptophan. Findings also revealed a meaningful link between hydrate formation and operating pressure observed for the exact temperature settings. High pressures facilitate the hydrate formation by reduced induction times with relatively higher formation rates, highlighting the subcooling effect on hydrate formation conditions. Overall, it can be concluded that using tryptophan as a biodegradable kinetic promotor considerably enhances the hydrate-based desalination process, making it more sustainable and cost-effective.

**Keywords:** amino acid; $CO_2$ hydrates; formation kinetics; hydrate-desalination; hydrate promotor





## 1. Introduction

Freshwater is essential for life, not only for human consumption and ecosystem support but also for industrial and agricultural purposes [1,2]. Water scarcity affects more than 80 countries, affecting about 40% of the world's population [2]. Moreover, due to unchecked population growth, decreased water quality has led to water scarcity [3,4]. Government agencies and municipal, regional, and international entities are developing programs to find new water sources and regulations to regulate energy and water demand to integrate present resources with population growth and industrial expansion [2,5–7].

Oceans are Earth's natural water reserves. Ice caps and other frozen formations safeguard 2% of the world's water supply, whereas only 0.5% is available freshwater. The mineral content of seawater is too high to use directly [8]. Thus, seawater desalination should be researched to meet rising freshwater demand using rich seawater resources. Desalination removes salts and minerals from water to make it drinkable. Seawater contains low amounts of $Na^+$, $Ca^{2+}$, $K^+$, $Mg^{2+}$, $(SO_4)^{2-}$, $Cl^-$, and other ions [8].

Recent desalination processes have greatly improved. Single-phase membrane techniques and phase change thermal methods dominate desalination [4,9,10]. Single-phase desalination uses membranes. Application of electro dialysis (ED) [11–13] and reverse osmosis (RO) [9,12,14–17] are prime examples of membrane-based desalination. Phase shift pathways evaporate saline water and produce fresh water using thermal energy from fossil fuels, solar energy, or nuclear energy. Phase shift desalination methods include solar distiller [4,18], vapor compression (VC) distillation [4,12,19], multi-effect (ME) distillation [9,18], and crystallization (hydrate freezing) [2,20–22]. These solutions work, but they have drawbacks. MSF uses energy, while RO requires capital investment and maintenance [9,19]. Both processes must discharge 50% of their input volume as a concentrate and cannot remove all impurities from saline or contaminated water [23]. These methods are too expensive for rural production [24]. Therefore, new technologies are needed to address these concerns. Electrodialysis–ion exchange desalination methods such as EDI are being studied. Membrane distillation (MD) and gas hydrate procedures are also innovative.

Clathrate hydrates, also known as gas hydrates, are non-stoichiometric solid inclusion compounds generated at high pressures and low temperatures when water molecules (host) encapsulate gas molecules (guest) via H-bonding [25–32]. Because of the size of the hydrate cages, which ranges from 0.395 to 0.571 nm [33,34], any dissolved ions and salts are isolated from the hydrate crystals during hydrate production [35]. The hydrate crystals formed are then removed from the brine solution to produce freshwater [34]. The latter is obtained by dissociating the hydrate crystals with heat or pressure, leaving behind the guest gas to be recycled in the process [2,34]. Gas hydrate-based desalination (GHBD) has a definite future alternative, and to expedite the hydrate formation, additives such as tetrahydro ferroan (THF), and cyclopentane (CP) shift the HLVE curve towards lower pressure and higher temperature region [36,37]. Similarly, the formation kinetics can be accelerated through the kinetic promotors, which drop the hydrate nucleation time and improve the rate of hydrate growth. Examples are sodium dodecyl benzene (SDS), amino acids, and ionic liquids [38,39]. Few studies have highlighted the combination of thermodynamic and kinetic promotors for facilitating the HLVE curve and kinetic promotion for hydrates [34,40,41].

In 2011, Park et al. [42] initially announced that they could generate potable water from gas hydrates 50% cheaper than current technologies. Javanmardi et al. [43] demonstrated that desalinating very saline water with the gas hydrate technique uses less energy and avoids pretreatment. A recent study of several desalination processes conducted by Montazeri and Kolliopoulos [34] indicated that high salt removal efficiency of greater than 90% can be achieved in hydrate-based desalination (HBD), with maximum salt rejection reported as 98.4. Likewise, the water recovery of the HBD processes has been estimated to be between 30 and 70 percent, which is significantly higher than any other traditional desalination approach. The schematic diagram of the HBD process is presented in Figure 1.

For hydrate-based desalination to be employed for household and drinking reasons, eco-friendly chemicals are necessary, highlighted by Montazeri and Kolliopoulos in their recent review on HBD. An ideal kinetic promoter for HBD should be non-toxic, inexpensive, recoverable, and ecologically acceptable, and possess high thermal conductivity and stability. By reducing the liquid–gas interfacial tension and increasing the solubility of gas hydrate formers, surfactants, for instance, have been found to reduce the induction time and increase the hydrate growth rate. Ionic liquids, amino acids, and biosurfactants have been the focus of recent research as hydrate promotors [26,44].

Biodegradable amino acids are essential ingredients in the human diet and have recently emerged as a highly effective class of KHPs [45,46]. Unlike surfactants (traditional

KHP molecules), they promise a clean medium of kinetic action, i.e., no foam formation. Although hydrophobic amino acids have been used for KHP prior, none of the previous studies highlighted the influence of aromatic side chain amino acids on HBD applications.

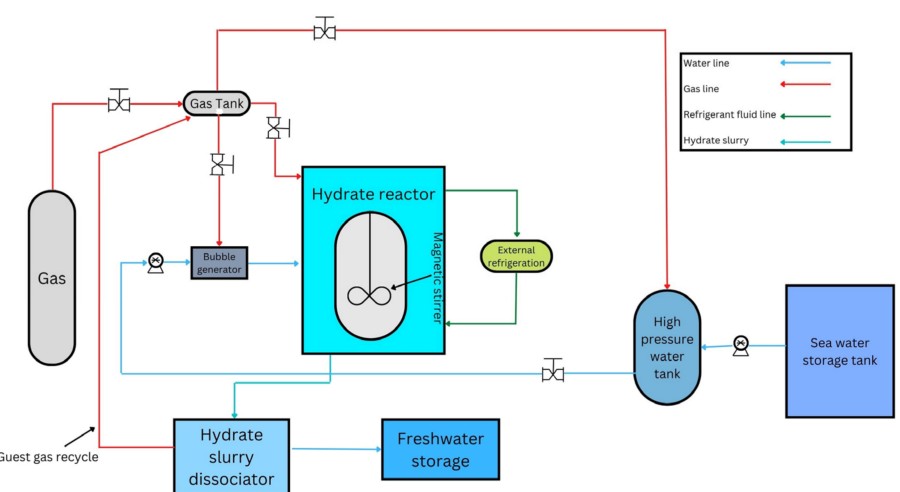

**Figure 1.** Schematic diagram of the hydrate-based desalination process.

The prime objective of this work is to evaluate the effect of biodegradable hydrophobic amino acid, i.e., tryptophan, on the $CO_2$ hydrate formation for designing a sustainable desalination process. In this regard, this research also focuses on assessing the impact of the driving force (subcooling and high pressure) on the $CO_2$ hydrate formation kinetics. Therefore, the present study is conducted on the kinetic hydrate performance of the natural amino acid tryptophan at varying concentrations (0.5, 1%, and 2%) with different pressure conditions (3.50 and 4.0 MPa) and temperatures (274.15, 275.15, 276.15, and 277.15 K) with $CO_2$ hydrates.

## 2. Materials and Methods

### 2.1. Materials

Table 1 presents the chemical and resources used in this study. L-Tryptophan (99 wt.%) and sodium chloride (99.5 wt.%) were purchased from Merck Millipore, Germany, while in-house deionized water was used to prepare the desired concentration aqueous L-Tryptophan and brine solutions. Moreover, the $CO_2$ gas was purchased from Linde Malaysia.

**Table 1.** List of chemicals used for this gas hydrate study.

| No. | Symbol | CAS Number | Chemical Label | Molecular Weight (g-mol$^{-1}$) | Purity | Supplier |
|-----|--------|------------|----------------|-------------------------------|--------|----------|
| 1 | $CO_2$ | 124-38-9 | Carbon dioxide gas | 44.01 | 99.99 mole % | Linde |
| 2 | $H_2O$ | 7732-18-5 | Water | 18.01 | Deionized | In-house |
| 3 | Tryptophan | 73-22-3 | L-Tryptophan | 204.23 | 99.0 wt% | Merck Millipore |
| 4 | NaCl | 7647-14-5 | Sodium Chloride | 58.44 | 99.5 wt% | Merck Millipore |

### 2.2. Methods

For all experimentation, a constant isochoric cooling system was employed to analyze the kinetic performance of desired brine and aqueous tryptophan solutions in a stainless-steel cell-equipped high-pressure reactor at high-pressure conditions. The high-pressure reactor is a jacketed high-pressure stainless-steel cell with an internal volume of 652.7 mL that is heated and cooled by an anti-freezing organic solvent solution (mono-ethylene glycol–ethanol) that circulates into the jacket via a thermostatic bath. A pressure-reducing valve connects it to a gas storage vessel. The pressure in the cell is measured with a 0–20 MPa

pressure sensor with a precision of 0.02 MPa, and the gas and liquid temperatures are measured with two PT100 probes with a precision of 0.1 K for both the gas and liquid phases. A magnetic rod-based agitator is installed in the reactor to provide efficient gas solubilization into the liquid phase and to improve thermal transfer. It rotates at the optimal speed of 400 RPM. It should be mentioned that the torque of this agitation mechanism is insufficient to stir any hydrate suspension. As a result, it is automatically stopped during the crystallization stage, and hydrate growth is always accomplished in quiescent or static conditions. Data were recorded on a laptop every 10 s using a custom-made Lab view$^{®}$ interface.

The reactor was first filled with a 100.0 0.1 cc volume of a liquid solution containing the brine and additions. The reactor was subsequently shut down, regulated at 284 and 282 K, much over the respective pressure's HLVE point, and flushed twice with $CO_2$ gas to remove the first air traces in the device. The gas input was then opened and promptly pressurized to the necessary pressure (3.5 or 4.0 MPa) to dissolve the gas in the solution. The amount of gas dissolved in the liquid was stabilized at that time (when the agitator was turned off). The reactor was then rapidly cooled from 284 K (at 4.0 MPa) to the appropriate operating temperature (274–277 K), and the system was kept at this temperature to create hydrates. An agitator is started during the cooling cycle, and data recording/logging is started for analysis reasons. The experiment was considered complete when no further temperature or pressure fluctuations were noticed. Figure 2 illustrates the combined schematic and experimental setup for understanding. The same system was used for earlier kinetic measurements; as a result, more details on the experimental setup and techniques can be found elsewhere [27,47–51].

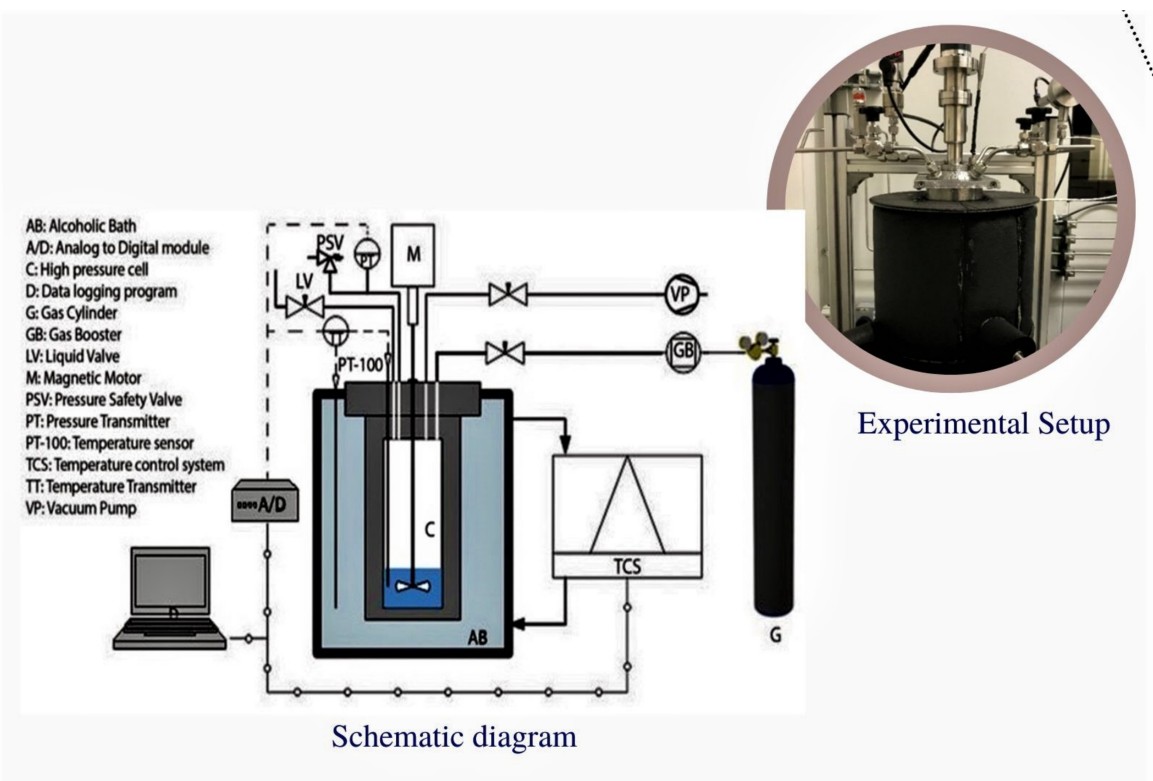

**Figure 2.** Depiction of combined schematic diagram and experimental setup used in the study.

The main kinetic hydrate evaluation parameters reported in this work are (i) induction time (the onset of hydrate formation), (ii) moles of $CO_2$ consumed (the amount of $CO_2$ gas consumed in the hydrate phase), (iii) rate of hydrate formation (for the first 30 min of hydrate formation), and (iv) water recovery (percentage) in the presence and absence

of tryptophan, which calculated using Equations (1)–(4) as effectively used in preceding studies [27,28,30,36,52,53].

$$t_{induction} = t_{hyd} - t_s \tag{1}$$

$$\Delta n_{CO_2} = \frac{V}{R}\left[\left(\frac{P}{zT}\right)_f - \left(\frac{P}{zT}\right)_o\right] \tag{2}$$

$$\frac{dn}{dt} = k_{30}(n_0 - n_{s30}) \tag{3}$$

$$\text{Water Recovery } C_w(\%) = \frac{\Delta n_H \times \text{hydration number}}{n_{H_2O}} \times 100 \tag{4}$$

where $nCO_2$ represents the moles of $CO_2$ consumed in the hydrate phase and $nH_2O$ represents the initial moles of free water. Similarly, z represents the compressibility factor of $CO_2$ at corresponding temperature and pressure conditions, V, R, T, and P, and $C_w$ reflects the volume of gas, the value of general gas law constant, experimental temperature, pressures, and water recovery ratios, respectively.

Moreover, Figures 3 and 4 describe the standard pressure–$CO_2$ hydrates moles consumed–time and temperature–$CO_2$ gas moles-time plot drawn for each experimental run for kinetic evaluation of different parameters. From Figure 3, the stirring start corresponds to time zero and is represented as $t_s$. When the agitating began, gas dissolved into the liquid phase until nucleation occurred after about 13.5 min, referred to as the induction time and considered the time from zero to the nucleation stage. Figure 4 highlights the temperature rise of 2.1 K and a sudden sharp drop in free $CO_2$ gas are the other indications used to detect hydrate nucleation. Following that, hydrate growth continued in two stages: (1) initial hydrate formation region, where catastrophic hydrate growth happened approximately for 30 min and is characterized as the initial formation rate. This was followed by a moderate hydrate development region with slight temperature and pressure changes until the experiment was completed [49,54]. All the kinetic experiments were run thrice, and the average value was considered for all the reported data to address the probabilistic nature of gas hydrate (crystallization) formation.

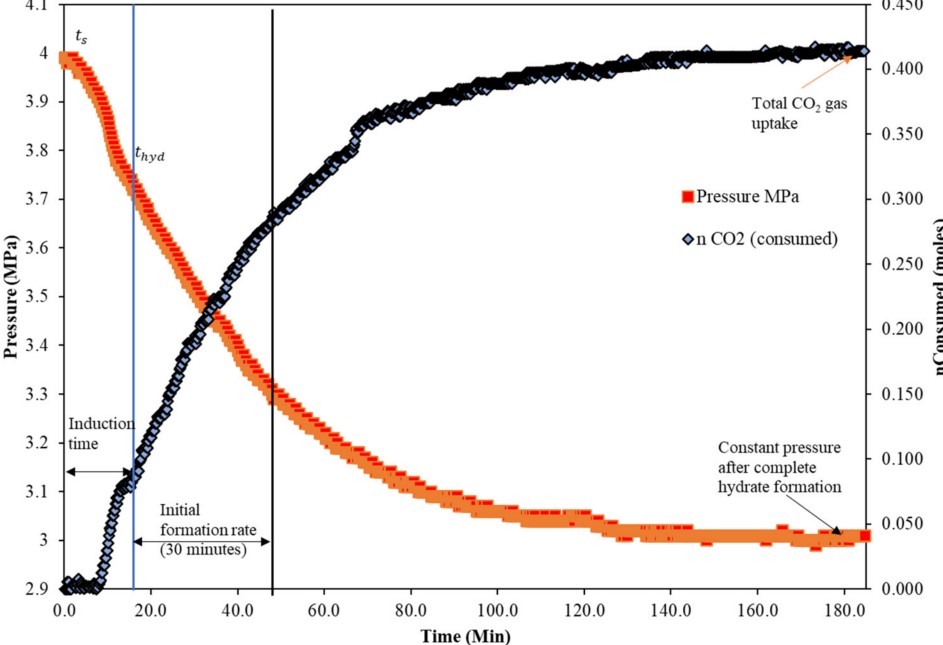

**Figure 3.** Kinetic measurement of aqueous brine solution at 277.0 K temperature in the presence of tryptophan using standard pressure–$CO_2$ hydrate mole consumption and time relationships in $CO_2$ gas–hydrate production.

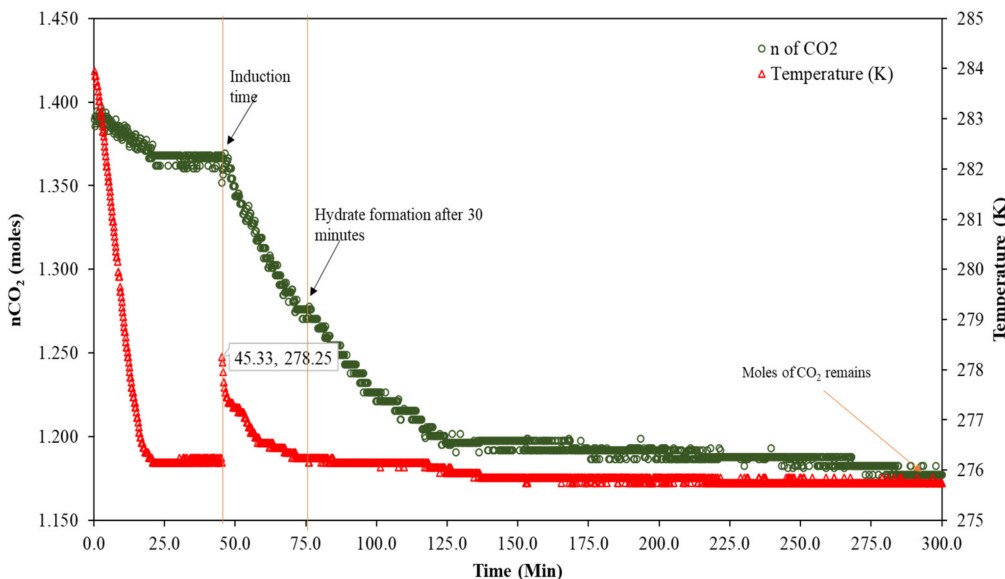

**Figure 4.** Kinetic measurement of aqueous brine solution at 276.0 K temperature using standard temperature–CO$_2$ gas (phase) moles and time relationships in CO$_2$ gas–hydrate production.

## 3. Results

The effect of 0.5, 1, and 2 wt.% tryptophan on the formation kinetics of CO$_2$ hydrate is investigated at two different pressures (3.5 and 4.0 MPa) and four different temperatures (274–277 K) for desalination application. These pressure and temperature parameters are chosen to obtain the most significant driving force for CO$_2$ gas.

### 3.1. Induction Time of CO$_2$ Hydrates in the Presence of Tryptophan

It is widely accepted that the induction time is an essential kinetic parameter to consider when evaluating the dynamics of hydrates formation. The induction delay of 0.5, 1, and 2 wt.% tryptophan–CO$_2$ hydrates is depicted at different experimental pressures in different parts of Figure 5. The finding from parts of Figure 5 clearly illustrated the influence of driving force (subcooling temperature and pressure) on induction time data in the presence of tryptophan. When we increase the experimental temperature from 274.15 to 277.15 for 1 wt.% tryptophan solution at 4.0 MPa condition, the induction time increases to about 94.6%, which is totally influenced by the subcooling condition.

On the other hand, the influence of tryptophan can be easily deduced by comparing the different concentrations of tryptophan (brine, 0.5, 1, and 2 wt.%) in the same temperature condition. For instance, if we evaluate at 4.0 MPa pressure and 276.15 K temperature conditions, the 0.5, 1, and 2 wt.% concentrations of tryptophan solutions can decrease the brine solution's induction time 52.6, 67.6, and 76.1%, respectively. Moreover, the influence of pressure is distinguished from the data analysis reported in Figure 5a,b. For further understanding, let us compare the induction time data of brine solution at 275.15 K temperature for both 3.5 MPa and 4.0 MPa conditions, where the lesser pressure condition (3.5 MPa) caused the 28.1% reduction in induction time indicated the influence of pressure for facilitation of hydrate formation. Overall, it was evident from the experimental findings that the higher subcooling and higher pressure conditions effectively facilitate the hydrate formations by reducing the induction time.

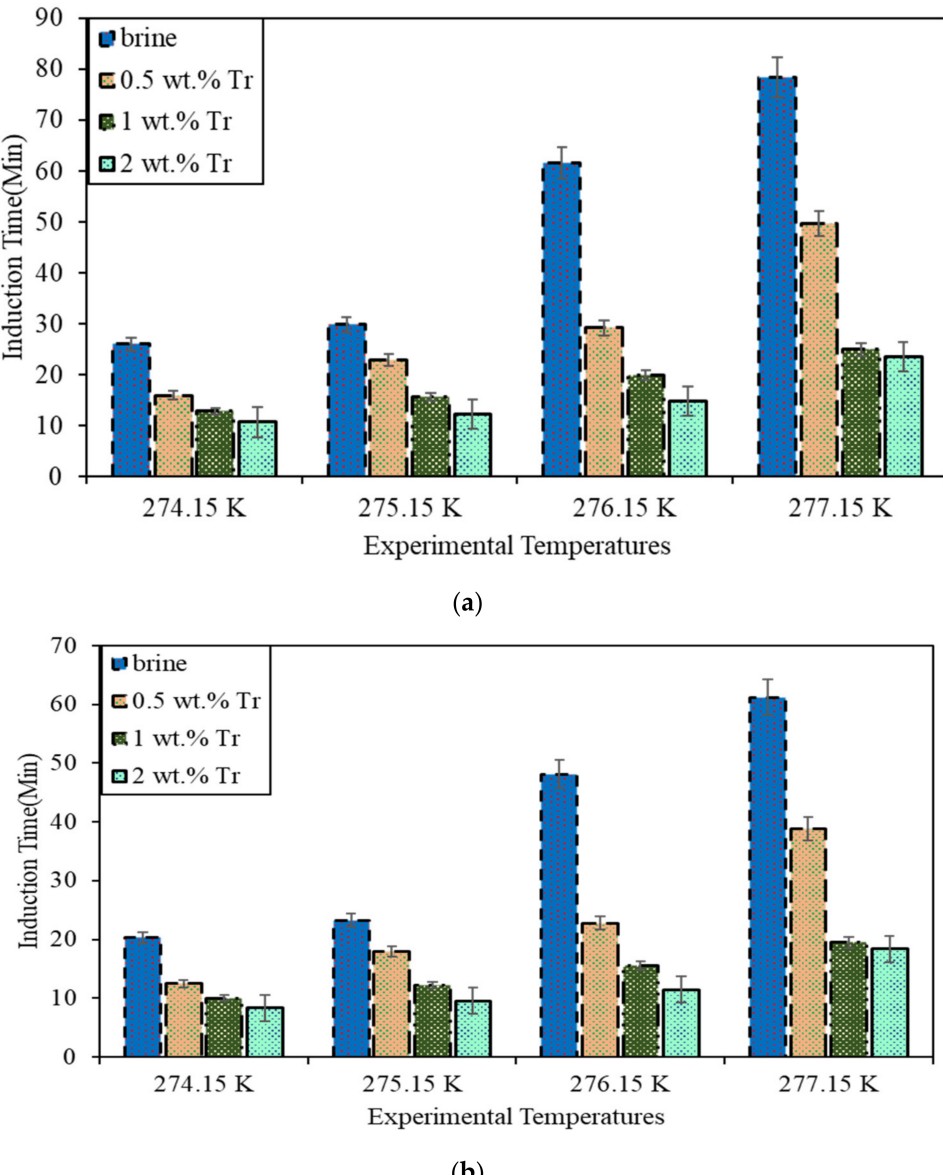

**Figure 5.** Effect of hydrate formation temperatures on the induction time of $CO_2$ hydrates for different tryptophan concentrations: (**a**) 3.5 MPa condition, (**b**) 4.0 MPa condition.

### 3.2. Initial Formation Rates of $CO_2$ Hydrates in the Presence of Tryptophan

Different parts of Figure 6a,b illustrate the moles of initial formation rates over a period of thirty minutes for aqueous tryptophan–$CO_2$ hydrates of 0.5, 1, and 2 wt.% at each of the four varied experimental temperatures of 274.15, 275.15, 276.15, and 277.0 K at different experimental pressures, respectively. The result from parts of Figure 6 clearly illustrated that the subcooling temperature and pressures have directly influenced the initial formation rates in the presence and absence of tryptophan due to induced driving force. For instance, when comparing the experimental temperatures of 274.15 and 277.15 for 1 wt.% tryptophan solution at 4.0 MPa condition, the initial formation rates considerably decrease to about 21.8%, which is influenced by the condensed driving force of subcooling in the system.

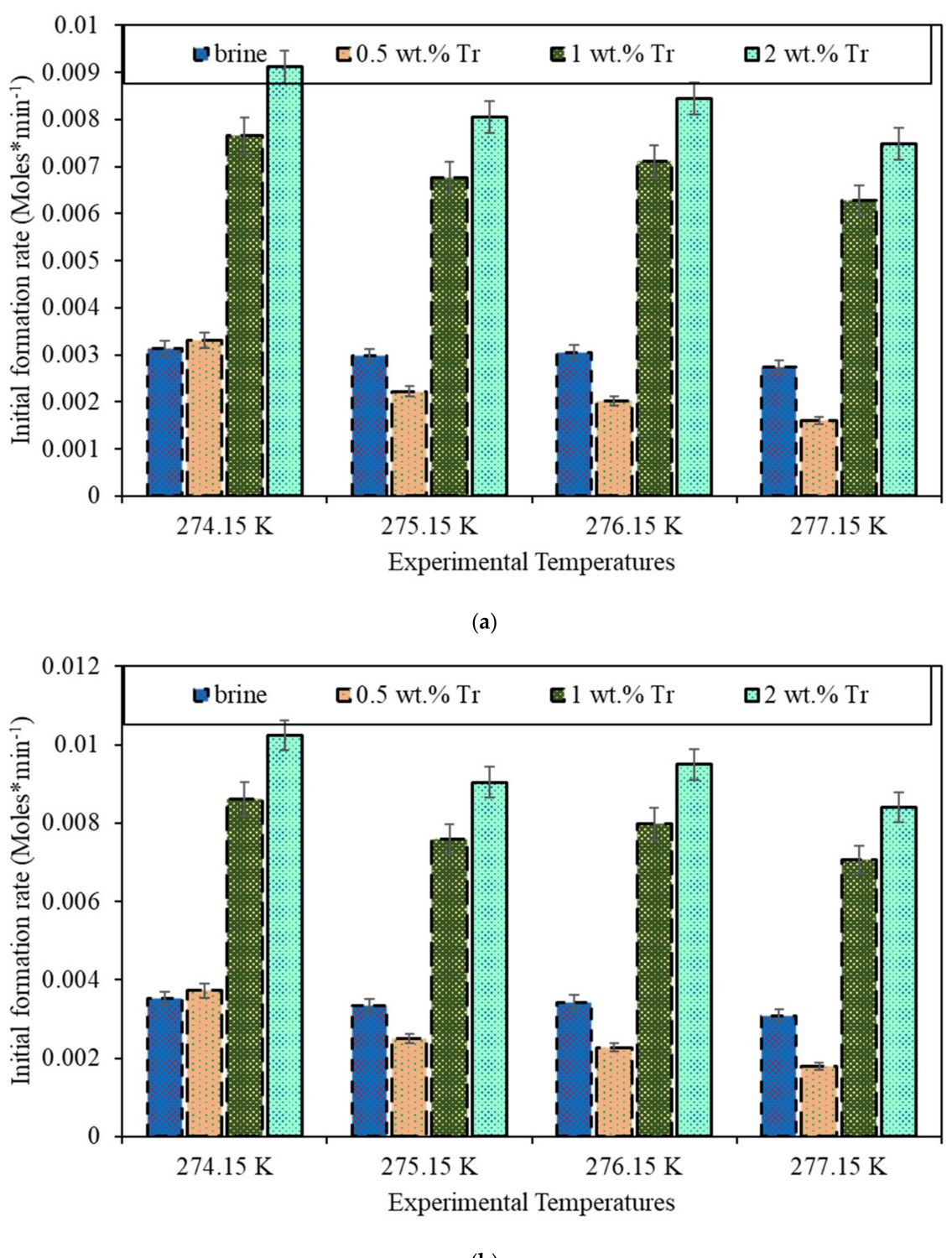

**Figure 6.** Effect of hydrate formation temperatures on the initial formation rates of $CO_2$ hydrates for different concentrations of tryptophan: (**a**) 3.5 MPa condition, (**b**) 4.0 MPa condition.

Conversely, the influence of tryptophan can be easily determined by comparison of different concentrations of tryptophan (1 and 2 wt.%) on the identical experimental temperature (276.15) and pressure (4.0 MPa) conditions. For instance, the 1 and 2 wt.% concentrations of tryptophan solutions increased the initial formation rates by about 150.7 and 176.6%, respectively. Moreover, the pressure effect can easily be distinguished from the data analysis represented in Figure 6a,b. For further understanding, let us compare the

initial formation rate data of 1 wt.% tryptophan solution at 275.15 K temperature for both 3.5 MPa and 4.0 MPa conditions. It was evident that the lesser pressure condition (3.5 MPa) caused about an 11% reduction in the initial formation rate highlighting the pressure effect on the facilitation of hydrate formation.

### 3.3. Gas Uptake of $CO_2$ Hydrates in the Presence of Tryptophan

Different parts of Figure 7a,b illustrate the overall moles of $CO_2$ gas uptake for aqueous tryptophan–$CO_2$ hydrates of 0.5, 1, and 2 wt.% at each of the four different experimental temperatures of 274.15, 275.15, 276.15, and 277.0 K at varied experimental pressures, respectively. Different parts of Figure 7 illustrate that the mole update of $CO_2$ gas is highly influenced by the subcooling temperatures and pressures in the absence and presence of tryptophan due to induced driving force. For example, we compare the 4.0 MPa pressure condition to the different experimental temperatures 274.15 and 277.15 in the presence of 1 wt.% tryptophan solution. The $CO_2$ gas consumption considerably decreased to about 2.1% due to reduced subcooling conditions providing a lesser driving force to the system. In contrast, the impact of tryptophan can be established by comparison of different concentrations of tryptophan (0.5, 1, and 2 wt.%) on the identical experimental temperature (276.15) and pressure (4.0 MPa) conditions. For instance, the 0.5, 1, and 2 wt.% concentrations of tryptophan solutions increased the $CO_2$ uptake by about 55.4, 73.0, and 123.2%, respectively.

Above and beyond, the pressure effect can be effectively observed from the data represented in Figure 7a,b. For further understanding, we have compared the $CO_2$ gas uptake values of 1 wt.% tryptophan solution at 275.15 K for both 3.5 MPa and 4.0 MPa conditions. It was evident that the lesser pressure (3.5 MPa) condition caused about a 14% reduction in the $CO_2$ gas uptake emphasizing the pressure effect on the acceleration of hydrate formation.

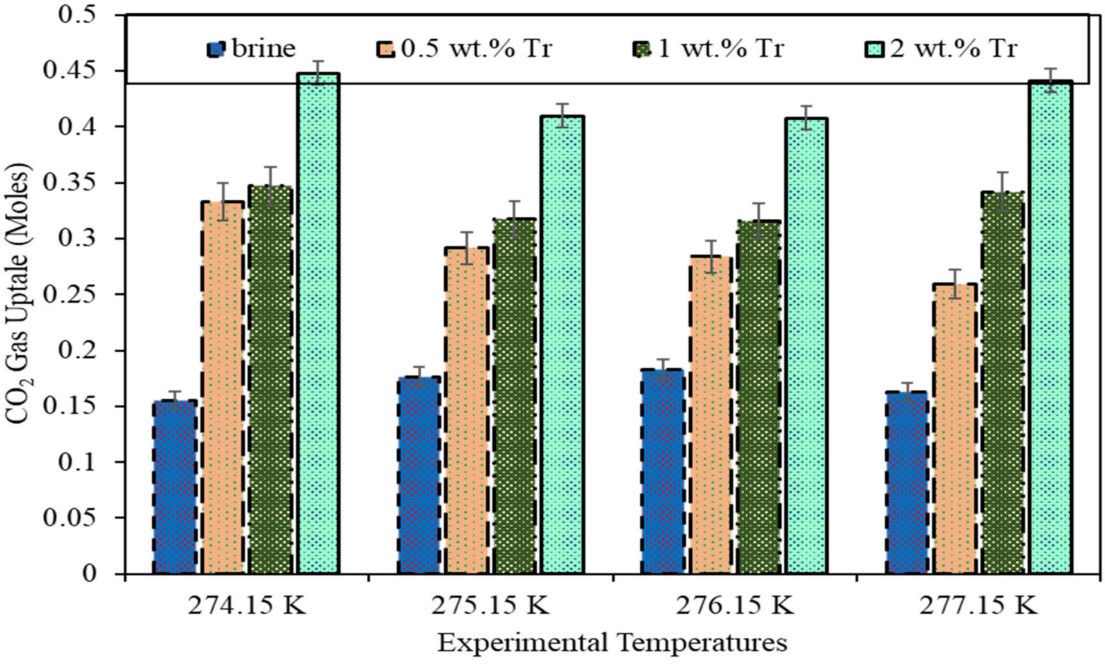

(a)

**Figure 7.** *Cont.*

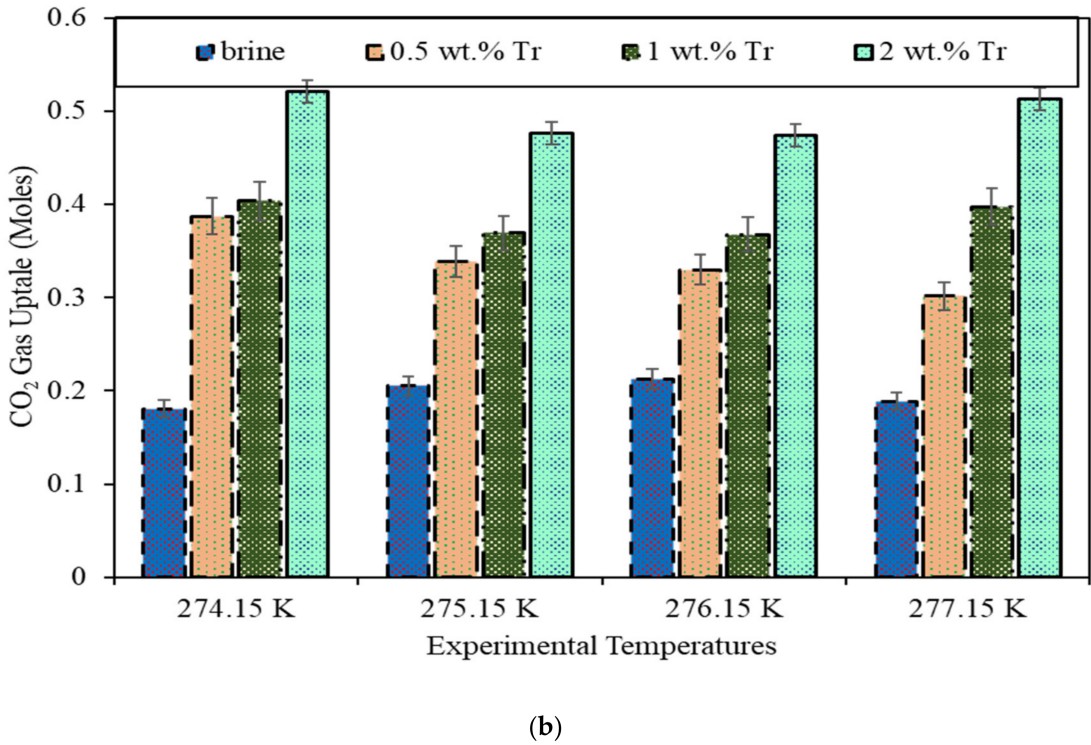

(**b**)

**Figure 7.** Effect of hydrate formation temperatures on the gas consumption of $CO_2$ hydrates for different concentrations of tryptophan: (**a**) 3.5 MPa condition, (**b**) 4.0 MPa condition.

### 3.4. Water Recovery from the $CO_2$ Hydrates in the Presence of Tryptophan

Different regions of Figure 8a,b demonstrate the water recovery (%) of brine, 0.5, 1, and 2 wt.% aqueous tryptophan–$CO_2$ hydrates for varying experimental temperatures (274.15, 275.15, 276.15, and 277.15 K) at 3.5 and 4.0 MPa, respectively. The result from parts of Figure 8 illustrated that the water recovery ratio is highly dependent on experimental pressure and subcooling temperatures and pressures. When comparing the experimental temperatures of 274.15 and 277.15 for 1 wt.% tryptophan solution at 4.0 MPa condition, the initial formation rates considerably decrease to about 3.1%, which is influenced by the reduced driving force in the system.

Conversely, the influence of tryptophan can be easily determined by comparison of different concentrations of tryptophan (0.5, 1, and 2 wt.%) on the identical experimental temperature (276.15) and pressure (4.0 MPa) conditions. For instance, the 0.5, 1, and 2 wt.% concentrations of tryptophan solutions increased the water recovery by about 55.1, 72.9, and 123.1%, respectively. Furthermore, the influence of pressure can be recognized from the data analysis represented in different parts of Figure 8a,b. For more interpretation, we compared the water recovery (%) data of 1 wt.% tryptophan solution at 275.15 K temperature for both 3.5 MPa and 4.0 MPa conditions. It was evident that the higher pressure condition (4.0 MPa) caused about a 16.2% increase in water recovery, highlighting the facilitation of hydrate formation of pressure due to the presence of a higher driving force.

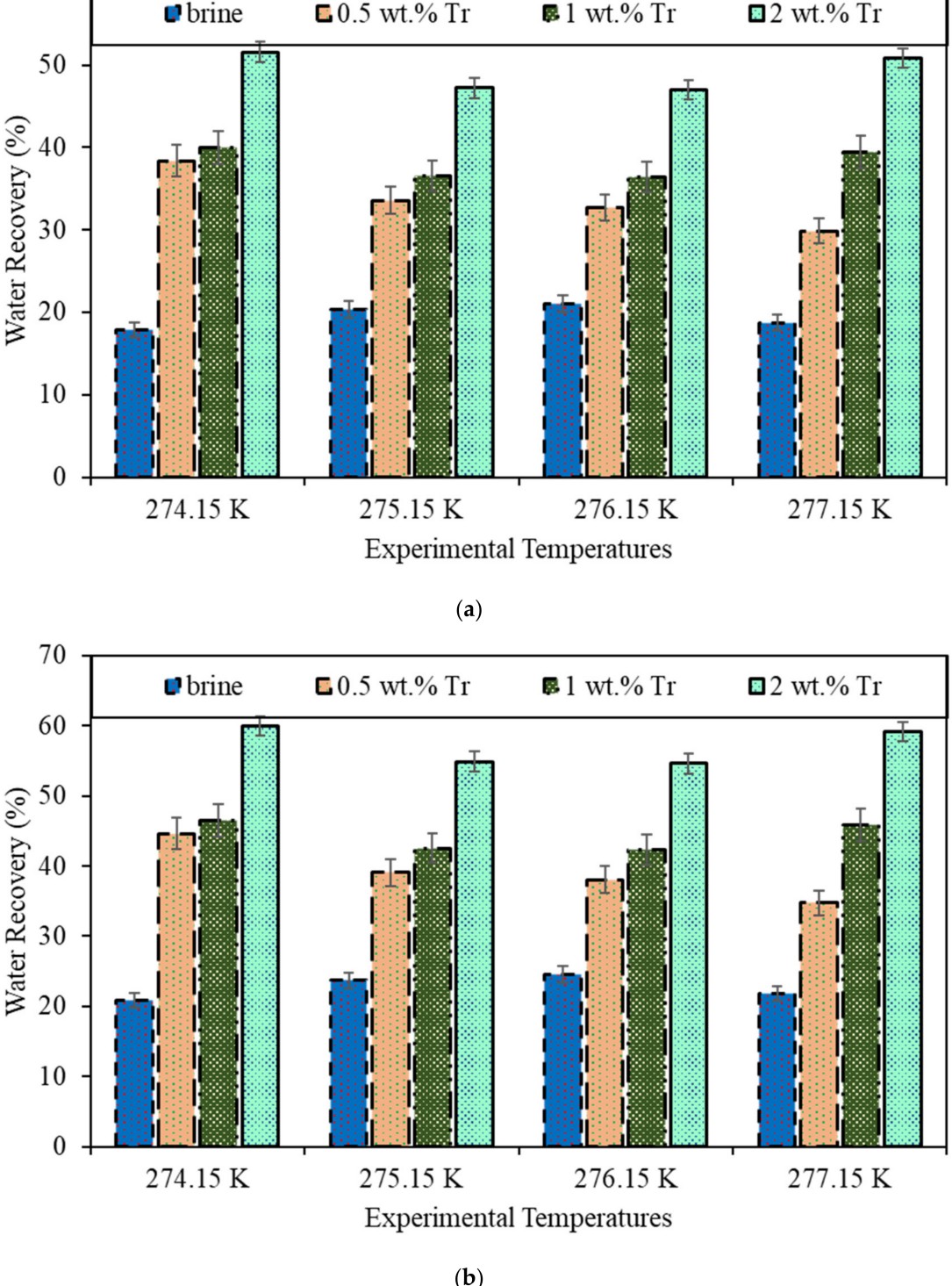

(**a**)

(**b**)

**Figure 8.** Effect of hydrate formation temperatures on the water recovery during $CO_2$ hydrates for different tryptophan concentrations: (**a**) 3.5 MPa condition, (**b**) 4.0 MPa condition.

## 4. Discussion

### 4.1. Effect of Tryptophan on the Induction Time of $CO_2$ Hydrates

According to the results observed in Figure 5b, induction time seems to be higher than the Figure 5a reported data for all reported experimental temperatures, highlighting the significance of higher experimental pressure for hydrate facilitations.

The effect of formation temperatures (sub-cooling) on induction time (hydrate delay) for $CO_2$ gas hydrates is accounted for in the results. Figure 5a,b confirmed that tryptophan successfully stimulates the formation of $CO_2$ gas hydrates under various subcooling conditions for both pressure conditions. The shortest induction times were observed for different tryptophan concentrations at the lowest temperature (274.15 K) condition due to a relatively more significant driving force in the system [51,55].

Moreover, the impact of concentration is also signified by the induction time data reported in Figure 5a,b. The higher concentration of tryptophan significantly reduces the induction time of $CO_2$ hydrates, indicating its mass transfer influence on hydrate formation and a comparatively more potent driving force.

### 4.2. Effect of Tryptophan on the Initial Formation Rate of $CO_2$ Hydrates

Initial formation rates findings revealed that all the examined tryptophan–$CO_2$ concentrations at different operating conditions promote hydrate formation by enhancing the hydrate formation rates more than the brine solution. The pressure effect significantly impacts the hydrate formation rate, as data reported in Figure 6b have higher formation rates than the corresponding low-pressure conditions reported in Figure 6a. For instance, in the brine solution at 274.15 K conditions, corresponding initial formation values were 0.00352 and 0.00313 for 4.0 and 3.5 MPa, respectively. Overall, the rate dropped by 11% when we varied experimental pressures due to the difference in the available driving force in the system [56,57]. Similar observations were evident for different concentrations of tryptophan where high-pressure conditions facilitate the hydrate formation and enhance the initial formation rates.

Like the induction time, the initial formation rates are also directly linked to the subcooling temperatures [50]. In contrast to pressure, it was found that the initial rates of hydrate formation decreased when the experimental temperatures increased (see Figure 6a,b). This is because there is a more potent driving force in place at lower experimental temperatures, which causes the rates of hydrate formation to increase.

### 4.3. Effect of Tryptophan on Gas Uptake of $CO_2$ Hydrates

The findings from different parts of Figure 7a,b exhibited that all tryptophan concentrations promoted $CO_2$ gas more than the brine solution, indicating robust kinetic promotion. The high-pressure condition (4.0 MPa) corresponds to higher $CO_2$ gas uptake than the lower-pressure condition (3.5 MPa) for all experimental temperatures. The finding also aligns with the preceding reported finding where tryptophan promotion performance enhanced with increased concentration due to added mass transfer and surface adsorption phenomenon [58]. The effect of subcooling conditions was further demonstrated by the fact that the moles of $CO_2$ gas uptake increased as subcooling conditions rose, notably for formations with a temperature of 274.15 K [27,51]. This may have been the outcome of the coexistence of a more vital driving force than high-temperature conditions.

### 4.4. Effect of Tryptophan on Water Recovery of $CO_2$ Hydrates

The finding from Figure 8a,b revealed that every concentration of tryptophan promotes water recovery more than the brine solution, which indicates its beneficial impact on hydrate kinetics. The effect of subcooling was negligible since different experimental temperatures demonstrated similar water recovery values at similar pressures. Different experimental pressure (3.5 and 4.0 MPa) reported in Figure 8a,b have a positive impact on the water recovery of the $CO_2$ hydrates. Due to the presence of the higher driving force at 4.0 MPa condition, the water recovery value was found to be 20.82% for brine solution at 274.15 K condition whereas corresponding 3.5 MPa conditions observe 17.90% water recovery value for brine solution, which shows a reduction of around 14% water recovery.

Similarly, adding tryptophan quantities in aqueous solutions enhances water recovery for all the experimental conditions (see Figure 8a,b). The lower concentration, i.e., 0.5 wt.% tryptophan, gave the lowest water recovery percentage in the case of aqueous tryptophan solutions due to the relatively slow mass transfer influence of the low concentration promoter

on hydrate development. Similarly, the increased quantities of tryptophan provide more surface adsorption and mass transfer ability in the system, resulting in a hydrate promotion capacity.

Overall, adding varying amounts of the biodegradable kinetic promoter tryptophan improved the formation kinetics of $CO_2$ hydrates by shortening the induction time and increasing gas absorption, hydrate formation rate, and water-to-hydrate conversion. Recently, Gaikwad et al. [59] achieved comparable results for higher (5.0 MPa) and lower (3.5 MPa) experimental pressures. They [59] reported that adding additional quantities of aqueous tryptophan resulted in a significant kinetic increase at both driving forces. However, understanding the mechanism of $CO_2$ hydrate production in the presence of tryptophan requires comprehensive morphological and molecular dynamics (MD) research. Veluswamy and coworkers [60] investigated leucine-induced methane hydrate production (an amino acid) morphology. They deduced that the hydrate generated for $CH_4$-leucine hydrates is porous and flexible. The increased porosity ness of $CH_4$-leucine hydrate aids capillary action. It was proposed that increasing the driving power of the bulk liquid (high pressure) resulted in greater hydrate formation.

The MD simulations of the $CH_4$-l-histidine system for kinetic hydrate promotion were validated by Bhattacharjee et al. [38]. Nguyen et al. [61] conducted another experimental and MD simulation research in the presence of hydrophobic (amino acid) surfaces. The discovery demonstrated that the coexistence of hydrophobic surfaces and interfacial gas enrichment caused water molecules to become spatially structured in the vicinity of hydrophobic surfaces. The phenomenon has been linked to the development gas hydrates in hydrophobic particle-containing environments [61]. Sa et al.'s investigation of various amino acids [62,63] revealed that amino acids have little or no influence on hydrate formation thermodynamics at lower concentrations of an amino acid (0.5 mol%). At larger concentrations (usually greater than 1 mol%), however, a dramatic shift in the thermodynamics of hydrate formation to more strict hydrate inhibition conditions is observed [62,63]. Therefore, it should be emphasized that all experiments in our current investigation were carried out at a concentration of 0.5 to 2 wt.% tryptophan. Hence, the chosen concentration will likely not affect the thermodynamics of hydrate formation and all the changes.

## 5. Conclusions

The current study investigated the kinetic behavior of a biodegradable hydrate promotor, tryptophan, in $CO_2$ hydrate systems for desalination application. Tryptophan–$CO_2$ hydrates were tested at different aqueous tryptophan concentrations with 1 wt.% brine solution for varied experimental temperatures and pressure conditions. The results showed that 1 wt.% tryptophan significantly improves the hydrate formation of $CO_2$ hydrates by reducing the induction period (49%), increasing the initial formation rates (144%), increasing $CO_2$ gas uptake (123%), and enhancing the water recovery at 274.15 K condition. The KHP influence is more effective with high subcooling circumstances, suggesting the driving force (subcooling temperatures) influencing the heat transfer phenomenon during hydrate formation. It was also evident that the increased quantities (0.5 to 2 wt.%) of tryptophan effectively promote hydrate nucleation and growth by disturbing water and gas dissolution activity through added adsorption and hydrophobic capabilities and potentially raising the sub-cooling temperature are possible strategies for promoting kinetic hydrate in all tested conditions. Therefore, the tryptophan formation kinetic data highlighted the importance of tryptophan as a biodegradable hydrate promotor that can be efficiently investigated in hydrate-based desalination systems.

**Author Contributions:** Conceptualization, M.S.K., B.L., and H.A.; methodology, M.S.K., B.L., and I.A. formal analysis, M.S.K., I.A., and H.A.; investigation, M.S.K. and B.L.; resources, B.L., A.M.S., and E.A.; data curation, M.S.K., I.A., and A.A.; writing—original draft preparation, M.S.K.; writing—review and editing, B.L., I.A., H.A., A.M.S., A.A., and O.B.; supervision, B.L.; project administration, B.L., H.A., and I.A.; funding acquisition, B.L., H.A., I.A., A.M.S., A.A., E.A., and O.B. All authors have read and agreed to the published version of the manuscript.

**Funding:** The Deputy Deanship of Research and Innovation, Ministry of Education in Saudi Arabia, grant number IFPNC−002-135-2020 and King Abdulaziz University DSR, Jeddah, Saudi Arabia, funded this research. The authors acknowledge using the $CO_2$ Research Center's (CO2RES) technical and laboratory facilities at Universiti Teknologi PETRONAS.

**Institutional Review Board Statement:** Not applicable.

**Informed Consent Statement:** Not applicable.

**Data Availability Statement:** Not applicable.

**Acknowledgments:** The authors would like to thank the Deputyship for Research and Innovation, Ministry of Education in Saudi Arabia, through project number IFPNC−002-135-2020, and King Abdulaziz University DSR, Jeddah, Saudi Arabia, for funding this research. The authors acknowledge using the $CO_2$ Research Center's (CO2RES) technical and laboratory facilities at Universiti Teknologi PETRONAS.

**Conflicts of Interest:** The authors declare no conflict of interest. All co-authors have seen and agree with the manuscript's contents, and there is no financial interest to report. We certify that the submission is original work and is not under review at any other publication.

## Nomenclature

| Abbreviation | Description | Abbreviation | Description |
| --- | --- | --- | --- |
| ED | Electro dialysis | $Cw$ | Water recovery ratios |
| VD | Vacuum distillation | T | Temperature (K) |
| V | The volume of the gas phase (mL) | P | Pressure (MPa) |
| R | Universal gas constant | SDS | Sodium dodecyl sulphate |
| z | Compressibility factor | KHP | Kinetic hydrate promotor |
| $CH_4$ | Methane | RO | Reverse osmosis |
| MSF | Multi-stage flash distillation | CP | Cyclopentane |
| MD | Membrane distillation | HBD | Hydrate based desalination |
| HLVE | Hydrate liquid–vapor equilibrium | GHBD | Gas hydrate-based desalination |
| $CO_2$ | Carbon dioxide | $H_2O$ | Water |
| MD | Molecular dynamics | NaCl | Sodium chloride |
| THF | Tetrahydro ferroan | | |

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
