# Peer review of "Formation Kinetics Evaluation for Designing Sustainable Carbon Dioxide-Based Hydrate Desalination via Tryptophan as a Biodegradable Hydrate Promotor"

_sustainability, doi:10.3390/su15010788_

Round 1

Reviewer 1 Report

1)        Quantitative results are suggested to be given in the abstract, not only the experimental conditions.

2)        The TDS of seawater mentioned in the introduction is wrong. 

3)        Too much contents in the introduction with little relationship with clathrate hydrate desalination. The authors should focus on the new desalination technology you used. The scarcity of freshwater is well-known. Current research focusing on hydrate desalination is suggested to be reviewed briefly for readers.

4)        The research objectives should be emphasized.

5)        Format of some chemicals should be noted, e.g., H2O, CO2.

6)        Experimental methods and conditions should be more detailed.

7)        The results section should not only show the readers the result figures. Actually, the authors described these results in the Discussion section, and discussion should indeed be strengthened. 

Author Response

Reviewer 1:

  • Quantitative results are suggested to be given in the abstract, not only the experimental conditions.

The authors are thankful for the respected reviewer’s comment, and the experimental findings are included in the abstract section, lines 26-32.

  • The TDS of seawater mentioned in the introduction is wrong.

The authors are thankful for the respected reviewer’s comment, the wrong TDS information is deleted from the introduction section.

  • Too much contents in the introduction with little relationship with clathrate hydrate desalination. The authors should focus on the new desalination technology you used. The scarcity of freshwater is well-known. Current research focusing on hydrate desalination is suggested to be reviewed briefly for readers.

The authors are thankful for the respected reviewer’s comment; the introduction section is curtailed, and undesired information is removed in the introduction of the revised manuscript. Moreover, more relevant information and literature have been added for hydrate desalination, lines 72-111.

  • The research objectives should be emphasized.

The research objective is highlighted in the revised manuscript's last paragraph of the introduction section, lines 112-115.

  • Format of some chemicals should be noted, e.g., H2O, CO2.

The format of the chemical is checked and updated in the revised manuscript. In addition, a nomenclature section is also added before the references section of the revised manuscript for easiness of the readers, line 355.

  • Experimental methods and conditions should be more detailed.

As requested by the respected reviewer, the method and experimental conditions are further elaborated in the revised manuscript, lines 133-157.

  • The results section should not only show the readers the result figures. Actually, the authors described these results in the Discussion section, and discussion should indeed be strengthened.

The revised manuscript's discussion section is enhanced and strengthened as requested by the respected reviewer, lines 288-315.

Reviewer 2 Report

The manuscript proposed a hydrate desalination process via tryptophan as a biodegradable hydrate promotor, and the formation kinetics was evaluated. Although the different concentrations of tryptophan, pressures, and the temperature of different hydrated carbon dioxide were investigated, I cannot find the innovation of this work. Also, there is no mechanism analysis of the formation kinetics mentioned before. So, I do not recommend the publication of this work. Some issues are listed below.

1.        Abstract is too long, and should be shorten.

2.        What is the innovation of this work? The innovation should be clarified in the section of introduction.

3.        It is better to give a schematic diagram for the hydrate-based desalination via tryptophan as a hydrate promotor.

4.        Line 101: what does the HBD mean?

5.        The language should be polished. For instance, Lines 109-113, “Therefore, ionic liquids, amino acids, and biosurfactants have been the focus of research. Therefore, the present study….”.

6.        The definition of Figure 1 is not high enough.

7.        Eqs. 1-4 are not clear, especially for the subscript.

8.        In the section of results and discussion, I cannot find some mechanisms of the formation kinetics. There are only many experiment results.

9.        I am confused why the section 6 patents added in the manuscript.

Author Response

Reviewer 2

The manuscript proposed a hydrate desalination process via tryptophan as a biodegradable hydrate promotor, and the formation kinetics was evaluated. Although the different concentrations of tryptophan, pressures, and the temperature of different hydrated carbon dioxide were investigated, I cannot find the innovation of this work. Also, there is no mechanism analysis of the formation kinetics mentioned before. So, I do not recommend the publication of this work. Some issues are listed below.

  1. Abstract is too long, and should be shorten.

The authors are thankful for the respected reviewer’s comment, and unwanted information is eliminated from the abstract section.

  1. What is the innovation of this work? The innovation should be clarified in the section of introduction.

The authors are thankful for the respected reviewer’s comment. The research objective and innovation are highlighted in the revised manuscript's last paragraph of the introduction section, lines 112-115.

  1. It is better to give a schematic diagram for the hydrate-based desalination via tryptophan as a hydrate promotor.

The authors are grateful for the respected reviewer’s valuable comment. The schematic diagram of the proposed hydrate-based desalination is added in the revised manuscript as Figure 1, lines 95-98.

  1. Line 101: what does the HBD mean?

The author meant to say hydrate-based desalination, which is added in the revised manuscript. In addition, a nomenclature section is also added before the references section of the revised manuscript for easiness of the readers, line 355.

  1. The language should be polished. For instance, Lines 109-113, “Therefore, ionic liquids, amino acids, and biosurfactants have been the focus of research. Therefore, the present study….”.

The authors are thankful for the valuable comments. The English language proofreading has been conducted, and grammatical and typo errors are updated from the revised manuscript.

  1. The definition of Figure 1 is not high enough.

The authors are thankful for the respected reviewer’s comments, and the description of the experimental procedure methods is updated in the revised manuscript, lines 133-157.

  1. Eqs. 1-4 are not clear, especially for the subscript.

       The authors are thankful for the reviewer’s comments, and the equations are updated in the revised manuscript, line 168.

  1. In the section of results and discussion, I cannot find some mechanisms of the formation kinetics. There are only many experiment results.

       The revised manuscript's discussion section is enhanced and strengthened as requested by the respected reviewer, lines 288-315.

  1. I am confused why the section 6 patents added in the manuscript.

The section was added due to the requirement from the publisher. However, the heading number was deleted from the revised manuscript, lines 333-334.

Reviewer 3 Report

The manuscript presents an experimental study on tryptophan as a hydrate promoter. The document is well-written and presents a good structure. Despite my positive comments, some aspects should be reviewed before publication:

a) In a desalination process proposal, how do you consider removing the CO2 absorbed (by solubility) in the water?

b) During the tests, how do you know that pressure equilibrium (Peq) has been reached at the evaluated temperature? Maybe by making pressure vs. temperature graphs, including HLV equilibrium data for the water-CO2 system, it is possible to check if equilibrium has been reached. Review previous studies: 10.1016/j.fuel.2014.01.025.

c) If the desalination of seawater is desired, which commonly contains 3.5 wt.% of salts, why is brine used at 1 wt.%? Please justify.

d) There is a question that arises when looking at Figure 6. If the effect of the promoter (tryptophan) is “only” kinetic, is it justified the gas uptake changes? Could it be that it also has a thermodynamic effect? One can consider that if the effect is only kinetic and the system reaches equilibrium, the final pressure must be the same at a given temperature. Review previous studies: 10.1016/j.ces.2016.06.034; 10.1016/j.cherd.2013.12.007.

Minor observation:

It is suggested to place a list of abbreviations at the end of the document.

Author Response

Reviewer 3:

The manuscript presents an experimental study on tryptophan as a hydrate promoter. The document is well-written and presents a good structure. Despite my positive comments, some aspects should be reviewed before publication:

a) In a desalination process proposal, how do you consider removing the CO2 absorbed (by solubility) in the water?

The authors are thankful for the valuable comment from the respected reviewer. The authors agree that the byproduct after hydrate desalination is desalinated water with little dissolved CO2 in it. The dissolved CO2 can be eliminated by post-treatment of fresh water through thermal decarbonization or activated carbon treatment. Please note that we have not considered this aspect in the current study. However, we will surely incorporate this point in our future work.

b) During the tests, how do you know that pressure equilibrium (Peq) has been reached at the evaluated temperature? Maybe by making pressure vs. temperature graphs, including HLV equilibrium data for the water-CO2 system, it is possible to check if equilibrium has been reached. Review previous studies: 10.1016/j.fuel.2014.01.025.

The authors are thankful for the reviewer’s comment. The present study is only focused on the formation kinetic evaluation of the low concentration of tryptophane, which is mainly influenced by the driving force. The thermodynamic influence mainly comes at larger concentrations (usually greater than 1 mol%). It should be noted that all experiments in our current investigation were carried out at a concentration of 0.5 to 2 wt.% tryptophan, which is less than 1 mol%. Hence, the chosen concentration will likely not affect the HLVE conditions of hydrate formation, and all the changes only occur due to kinetic influence. The related paper is cited in the text and relevant discussion,n is added in the revised manuscript, lines 307-315.

c) If the desalination of seawater is desired, which commonly contains 3.5 wt.% of salts, why is brine used at 1 wt.%? Please justify.

The authors are thankful for respected reviews for insightful comments. Our current project emphasized evaluating different brine concentrations for the desalination process. So, initially, we evaluated 1 wt.% of brine solution, and we already proposed to evaluate 2, 2.5, 3 and 3.5 wt.% brine solutions for our ongoing project. 

d) There is a question that arises when looking at Figure 6. If the effect of the promoter (tryptophan) is “only” kinetic, is it justified that the gas uptake changes? Could it be that it also has a thermodynamic effect? One can consider that if the effect is only kinetic and the system reaches equilibrium, the final pressure must be the same at a given temperature. Review previous studies: 10.1016/j.ces.2016.06.034; 10.1016/j.cherd.2013.12.007.

The author is thankful to the respected reviewer for their valuable input. The thermodynamic influence is generally considered based on the changes observed in HLVE of CO2 hydrate in the presence of the additive, which usually happens at a higher concentration that influences the hydrogen bonding network of water molecules. As mentioned in the revised manuscript, lines 307-313. The selected compositions (0.5-2 wt.%) are in very fewer quantities to disrupt the hydrogen bonding network of bulk water molecules hence little to no thermodynamic influence is attributed. During the kinetic experiment, as mentioned in Figure 3 of the revised manuscript, the initial pressure of 4.0 MPa indicated only liquid and gas phase present in the reactor since experiments are initiated above the HLVE condition. The sudden pressure dropped, and simultaneous gas uptake indicated that the CO2 gas encapsulation in the hydrate structure initially happens at the constant operating temperature of 276 K (see Figure 4). The sharp temperature rise of the reactor temperature from 276 K to 278.25 K indicates the exothermic nature of the hydrate formation. Once the bulk hydrate formation is completed, the reactor again attains the set temperature of 276 K. At the same time, the gas phase pressure is reduced due to the formation of the hydrate phase in the reactor, which we analyzed as kinetic formation data. The final pressure of the gas phase differs because the difference has resulted in the form of CO2 hydrate. The related references are cited in the text, line 85.

Minor observation:

It is suggested to place a list of abbreviations at the end of the document.

As suggested by the respected reviewer, a nomenclature section is added in the revised manuscript before the references section for easiness of the readers, line 355.

Round 2

Reviewer 1 Report

The authors have responded my comments and improved the manuscript, but the results and discussion structure of the paper is still questioned. The authors only show the figures in the Results session, without any analysis related to the single experimental result. The Discussion session should talk about the relations or mechanisms that cannot be discussed in only one experiment or one figure.

And the subtitle of 3.3 is duplicated.

Author Response

1)The authors have responded my comments and improved the manuscript, but the results and discussion structure of the paper is still questioned. The authors only show the figures in the Results session, without any analysis related to the single experimental result. The Discussion session should talk about the relations or mechanisms that cannot be discussed in only one experiment or one figure.

  • The authors are thankful for the respected reviewer's comment, the further description and interpretation of all the reported results are added in the revised manuscript, lines 198-218, 213-239, 247-263 and 270-286.

2)And the subtitle of 3.3 is duplicated.

  • The authors are grateful for the respected reviewer's comment; the mistake was rectified in the revised manuscript, line 267.

Reviewer 2 Report

The manuscript has been revised well, but some mistakes should be addressed before publication.

1.        Line 55: (SO4)- should be revised as SO42-. Besides, Ca++ and Mg++ should be revised as Ca2+ and Mg2+, respectively.

2.        References cited in the revised manuscript should be checked carefully, because I find some incorrect citations. For example, in Line 58, the ref 12 is not a study referring to electrodialysis, which can be replaced by the ref Journal of Membrane Science 570–571 (2019) 245–257. In addition, the authors cite excessive references to support one sentence in many parts of the manuscript, I suggest the authors should cite the most relevant references to support the corresponding sentence.

3.        In Fig. 4, For “45.33333333, 278.25”, the decimal points should be revised and keep one or two decimal places.

4.        For an article, I think the section of “Patents This US patent application is submitted based on the work reported in this manuscript” should not appear in the manuscript. So, please delete this section.

Author Response

The manuscript has been revised well, but some mistakes should be addressed before publication.

  1. Line 55: (SO4)- should be revised as SO42-. Besides, Ca++ and Mg++ should be revised as Ca2+ and Mg2+, respectively.
  • The authors are grateful for the respected reviewer's comment; the typo errors were rectified in the revised manuscript, line 55.
  1. References cited in the revised manuscript should be checked carefully, because I find some incorrect citations. For example, in Line 58, the ref 12 is not a study referring to electrodialysis, which can be replaced by the ref Journal of Membrane Science 570–571 (2019) 245–257. In addition, the authors cite excessive references to support one sentence in many parts of the manuscript, I suggest the authors should cite the most relevant references to support the corresponding sentence.
  • The authors are thankful for the respected reviewer's comment. The relevant reference is updated as requested by the respected reviewer, line 58. Furthermore, the other reference is also checked and updated where required.
  1. In Fig. 4, For "45.33333333, 278.25", the decimal points should be revised and keep one or two decimal places.
  • The authors are obliged to the respected reviewer's comment; Figure 4 is updated in the revised manuscript, line 188.
  1. For an article, I think the section of "Patents This US patent application is submitted based on the work reported in this manuscript" should not appear in the manuscript. So, please delete this section.
  • As requested by the respected reviewer, the patent section is removed from the revised manuscript.
